# Effects of Epstein-Barr Virus Infection on the Risk and Prognosis of Primary Laryngeal Squamous Cell Carcinoma: A Hospital-Based Case-Control Study in Taiwan

**DOI:** 10.3390/cancers13071741

**Published:** 2021-04-06

**Authors:** Li-Ang Lee, Tuan-Jen Fang, Hsueh-Yu Li, Hai-Hua Chuang, Chung-Jan Kang, Kai-Ping Chang, Chun-Ta Liao, Tse-Ching Chen, Chung-Guei Huang, Tzu-Chen Yen

**Affiliations:** 1Department of Otorhinolaryngology—Head and Neck Surgery, Chang Gung Memorial Hospital, Linkou Main Branch, Taoyuan City 33305, Taiwan; 5738@cgmh.org.tw (L.-A.L.); fang3109@cgmh.org.tw (T.-J.F.); hyli38@cgmh.org.tw (H.-Y.L.); keny@cgmh.org.tw (C.-J.K.); changkp@cgmh.org.tw (K.-P.C.); liaoct@cgmh.org.tw (C.-T.L.); 2Faculty of Medicine, Chang Gung University, Taoyuan City 33302, Taiwan; chhaihua@cgmh.org.tw (H.-H.C.); ctc323@cgmh.org.tw (T.-C.C.); 3Department of Family Medicine, Chang Gung Memorial Hospital, Linkou Main Branch, Taoyuan 33305, Taiwan; 4Department of Industrial Engineering and Management, National Taipei University of Technology, Taipei 10608, Taiwan; 5Department of Pathology, Chang Gung Memorial Hospital, Linkou Main Branch, Taoyuan City 33305, Taiwan; 6Department of Laboratory Medicine, Chang Gung Memorial Hospital, Linkou Main Branch, Taoyuan City 33305, Taiwan; 7Research Center for Emerging Viral Infections, Graduate Institute of Biomedical Sciences, Chang Gung University, Taoyuan City 33302, Taiwan; 8Department of Nuclear Medicine and Molecular Imaging Center, Chang Gung Memorial Hospital, Linkou Main Branch, Taoyuan City 33305, Taiwan

**Keywords:** *BCL-2*, *B2M*, *CD3*, *CD161*, Epstein–Barr virus, EBER, laryngeal squamous cell carcinoma, local recurrence, disease-free survival, risk

## Abstract

**Simple Summary:**

Epstein–Barr virus DNA positivity, age ≥ 55 years, cigarette smoking, and high *BCL-2*, *B2M*, and *CD161* expression were identified as independent risk factors for primary laryngeal squamous cell carcinoma. A high EBER signal and low CD3 expression significantly and independently predicted local recurrence and disease-free survival within five years. The information obtained in this study improves our understanding of viral infections in laryngeal cancer, and may guide future prevention, treatment, and follow-up strategies.

**Abstract:**

Mounting molecular evidence supports Epstein–Barr virus (EBV) involvement in the pathogenesis of laryngeal squamous cell carcinoma (LSCC); however, the epidemiological data are inconsistent. In this retrospective case-control study, we aimed to determine whether EBV infection underlies the risk and prognosis of LSCC. The prevalence of EBV infection, as analyzed using an EBV DNA polymerase chain reaction assay, was significantly higher in 42 Taiwanese patients with newly diagnosed primary LSCC, compared to 39 age- and sex-matched control patients without cancer (48% vs. 19%). Furthermore, most of the *EBER* signals detected using in situ hybridization were localized to the nuclei of tumor-infiltrating lymphocytes. In multivariate analysis, EBV DNA positivity, age ≥ 55 years, cigarette smoking, and high *BCL-2*, *B2M*, and *CD161* expression (assessed using immunohistochemistry) were identified as independent risk factors for LSCC. Furthermore, five-year local recurrence and disease-free survival rates were 34% and 58%, respectively, with a high *EBER* signal and low *CD3* expression independently predicting five-year local recurrence and disease-free survival. Our comprehensive profiling data accurately identified patients at risk for LSCC development, local recurrence, or disease-free survival. The information obtained in this study improves our understanding of EBV infection in LSCC, and may guide precision medicine for patients with LSCC.

## 1. Introduction

The long-term survival of patients with laryngeal squamous cell carcinoma (LSCC) is excellent when locoregional recurrence is controlled [1]. However, approximately 2.0% of deaths per annum are attributable to LSCC, with survivors frequently experiencing impaired voice quality [2,3]. Therefore, the etiologic risk factors for primary and recurrent LSCC need to be thoroughly studied; potential risk factors include cigarette smoking, alcohol consumption, environmental carcinogens, and gene mutations [4]. According to previous studies, additional factors include human papillomavirus [4,5,6], contrary to our findings [7]; and Epstein–Barr virus (EBV) [8,9,10,11,12], contrary to various reports [5,13,14]. 

As the upper aerodigestive tract is in constant contact with the environment, it is easily infected by EBV, making it susceptible to local carcinogenesis [9]

In nasopharyngeal carcinoma, local infiltration of EBV-infected leukocytes creates a tumor microenvironment that promotes tumor development [15]. However, the effects of EBV infection on the host’s genetic susceptibility (e.g., expression of *BCL-2* (the Akt signaling pathway) [16], *MYC* (the TGF-β signaling pathway) [17], and *p16INK4a* (the cell cycle G1/S checkpoint signaling pathway) [7]) and immune response (e.g., expression of *B2M* [18], *CD3* [19], and *CD161* [20]) to LSCC have not been reported in detail. Moreover, the relationship between viral infections and LSCC development and prognosis requires further investigation. 

In this study, we aim to examine the following hypotheses: (1) EBV infection underlies the risk of primary LSCC, and (2) EBV infection affects the prognosis of LSCC, using a more comprehensive approach. The potential mechanisms of virus-related carcinogenesis will also be discussed. 

## 2. Materials and Methods

### 2.1. Study Populations

This was a retrospective case-control study. We included patients with laryngeal lesions who had undergone laryngeal surgery in the Department of Otorhinolaryngology-Head and Neck Surgery at Chang Gung Memorial Hospital, Linkou Main Branch (Taoyuan City, Taiwan) between 1 August 2012 and 31 December 2015. These patients had been recruited to a previous study exploring tumor microbiology in laryngeal cancer (funder: Chang Gung Medical Foundation; no. CMRPG3C0661–3). The primary inclusion criteria were as follows: age > 18 years, pathological diagnosis of a laryngeal lesion, and willingness to sign informed consent forms. The exclusion criteria included an unwillingness to undergo tissue examination and/or answer subjective questionnaires. All investigations were conducted in accordance with the World Medical Association Declaration of Helsinki. This study followed the recommendations for prognostic studies of tumor biomarkers (REMARK) [21].

In this study, the case group consisted of patients with newly diagnosed, histologically confirmed LSCC. The control group consisted of patients without a history of cancer who were treated for a non-malignant lesion of the larynx (NMLL), such as polyps and leukoplakia. Patients with initial NMLLs who had malignant transformations during the study period were excluded from this study. Case patients were matched to control patients according to sex.

This study also aimed to investigate the effects of EBV infection on long-term prognoses, including five-year local recurrence (LR), neck recurrence (NR), distant metastasis (DM), disease-free survival (DFS), disease-specific survival (DSS), and overall survival (OS), of primary LSCCs. Medical records obtained from the electronic health information system until 28 February 2021 were reviewed. The study was approved by the Institutional Review Board (no. 202100292B0) of the Chang Gung Medical Foundation (Taoyuan City, Taiwan). Written informed consent for this study was waived.

### 2.2. Clinicopathologic Evaluations

Data on the following variables were collected at enrollment: age at diagnosis, sex, smoking status, alcohol consumption status, pathological diagnosis, pathological staging (according to the 2009 revision of the American Joint Committee on Cancer tumor-node-metastasis staging system) [22], treatment modality, and LR. LSCC patients underwent transoral laser microsurgery (TLM), radiation therapy (RT), or concurrent chemoradiotherapy (CCRT) for T1–3 tumors, or total laryngectomy with neck dissection for T4a tumors, as previously described [3,23,24]. *En bloc* resection was attempted for all lesions, and tumor-free margins were confirmed using frozen-section analysis during the operation [3,23,24]. If positive primary excision margins were noted, additional extended resections were performed until all secondary margins were negative for malignancy [25]. Patients for whom complete resection was not possible, as determined by a multidisciplinary tumor board at the hospital, were suggested for adjuvant RT or CCRT [23].

In this study, LR was defined as a positive biopsy in the region of the primary tumor following a complete response to treatment, or in the area of the primary tumor following radical surgery (i.e., a negative post-treatment screen). NR was defined as a positive cytology/biopsy in the cervical lymphatic region following primary surgery. DM was identified through biopsy or imaging, as verified by our tumor board. Therefore, the five-year LR, NR, DM, DFS, DSS, and OS rates were calculated.

### 2.3. Detection of EBV DNA in Tumor Tissue and Plasma

Formalin-fixed paraffin-embedded (FFPE) tissue sections were prepared from specimens that had been histologically evaluated for lesion type and tissue adequacy (≥10.0% LSCC/NMLL cells) using hematoxylin and eosin staining [26]. We have previously detailed the anti-contamination procedures elsewhere [27]. Three 5-μm-thick FFPE sections per specimen were used for DNA extraction (Lab Turbo Virus Mini Kit; TaiGen Biotechnology Co., Taipei, Taiwan) [7].

Plasma DNA from 10 mL of peripheral blood was extracted using a QIAamp DNA Blood MiniKit (Qiagen Inc., Valencia, CA, USA). Approximately 500 μL of each sample per column was eluted with 80 μL of distilled water and used for DNA detection [28].

Real-time polymerase chain reaction (PCR) assays were performed using an ABI Prism 7700 Sequence Detection Analyzer (Applied Biosystems, Carlsbad, CA, USA) to detect the *Bam*HI W fragment in the EBV genome; we previously detailed the real-time PCR protocol elsewhere [28]. Briefly, we used primers to detect the *Bam*HI W region sequence (forward: 5′CCCAACACTCCACCACACC3′; reverse: 5′TCTTAGGAGCTGTCCGAGGG3′; and a dual fluorescence-labeled oligomer: 5′[FAM]CACACACTACACACACCCACCCG TCTC[TAMRA]3′) [29]. EBV DNA positivity was defined as a detectable EBV-DNA signal after 40 cycles of PCR, whereas EBV DNA negativity was defined as a copy number of zero [30].

### 2.4. Detection of EBV Antibody in Serum 

Serum obtained from 3 mL of peripheral blood was used to determine serum EBV titers. An immunofluorescence assay specific for EBV viral capsid antigen (VCA) IgA (Meridian Bioscience Inc., Cincinnati, OH, USA) was performed in accordance with the manufacturer’s protocol. We have previously outlined the immunofluorescence assay protocol elsewhere [31]. Briefly, acetone-fixed B95.8 cell glass slides were applied to dilutions of patient sera. Appropriate concentrations of fluorescein isothiocyanate-conjugated anti-human IgA were applied following incubation and washing. The antibody titer of the test serum was defined as the reciprocal of the highest dilution that clearly showed apple-green fluorescence in 20% of cells. The cut-off values for EBV VCA IgA were set at 1:40 [31].

### 2.5. Tissue Microarray Construction 

All routine hematoxylin and eosin-stained sections of FFPE tissue blocks that had been histologically evaluated for lesion type and tumor tissue adequacy were used for tissue microarray (TMA) construction, the protocol of which is described elsewhere [7,25]. Briefly, two regions within LSCC or NMLL foci per case were marked on hematoxylin and eosin-stained slides by a pathologist (TCC) blinded to the patient’s clinical information and used to assemble the recipient blocks. Tissue biopsy cores, each 1.0 mm in diameter and 4 µm in thickness, were taken from corresponding donor blocks. 

### 2.6. In Situ Hybridization

For in situ hybridization (ISH) detection of EBV-encoded small RNAs (*EBERs*), biotin-labeled EBV probe solutions (Bond Ready-to-Use EBER Probe, cat. # PB0589; Leica Biosystems Ltd., Newcastle, UK) were applied to TMA sections using an automated immunostainer (Bond-Max; Leica Microsystems GmbH, Wetzlar, Germany), according to the manufacturer’s instructions. Tumor tissue from EBV-positive nasopharyngeal carcinoma (NPC) was used as a positive control, while LSCC tissue from EBV-negative samples was used as a negative control. As *EBERs* are expressed in tumor-infiltrating lymphocytes (TILs), neighboring lymphocytes, and tumor cells [32], *EBER* ISH was considered “positive” when *EBER* staining was present in tumor cells and/or peritumoral stroma.

### 2.7. Immunohistochemistry Staining

Immunohistochemistry (IHC) was used to detect the expression of *BCL-2* (cat. # M088701; Dako North America, Inc., Carpinteria, CA, USA), *MYC* (cat. # Z2258; Zeta Corp., Sierra Madre, CA, USA), *p16INK4a* (cat. # E6H4; Roche Diagnostics GmbH, Heidelberg, Germany), *B2M* (cat. # NCL-B2MP; Leica Biosystems Ltd., Newcastle, UK), *CD3* (cat. #s NCL-L-CD3-565; Leica Biosystems Ltd., Newcastle, UK), and *CD161* (cat. # orb5815; Bioorbyt, Cambridge, UK). TMA tissue sections were stained using the same immunostains noted above, according to the manufacturer’s instructions. Detailed IHC procedures have been previously described [7,25]. For optimal detection, antibodies were diluted at the following ratios: 1:10 (*p16INK4a*), 1:500 (*BCL-2*), 1:50 (*MYC*), 1:100 (*B2M*), 1:200 (*CD3*), and 1:100 (*CD161*). Antibody reactions were performed at room temperature for 20 min.

### 2.8. Computer-Supported Evaluation 

Stained slides were digitalized at 40× magnification using an Aperio ScanScope scanner (Leica Biosystems, Richmond, IL, USA). Expression levels of *EBER* and histological marker proteins were digitally assessed in four specified regions of interest (ROIs), which contained the laryngeal mucosal lesion and submucosal tissue, using semi-quantitative image analysis (Tissue Studio v2.1; Definiens AG, Münich, Germany) [7,25,33]. The image analysis software allowed for the specific mining of staining intensities in the cell nuclei (*EBER* and *MYC*) or whole cells (*p16INK4a*, *BCL-2*, *B2M*, *CD3*, and *CD161*); nuclei and cells were categorized as “negative” or “positive” (weak, moderate, or high staining intensity). The positivity index (PI; %) ((number of positively stained cells or nuclei/total number of cells or nuclei) × 100) was calculated. The same investigator who was blinded to the clinical outcomes chose the ROIs and performed a computer-supported evaluation. 

### 2.9. Sample Size Calculation

Sample sizes were estimated according to our prior study (Fisher’s exact test, two-tailed α = 0.05, power = 0.80; allocation ratio = 1), which was performed in a previous study investigating EBV infections in patients with LSCC (52%), vocal cord leukoplakia, and polyps (20%). A total of at least 80 patients were required to achieve the aims of the study. 

### 2.10. Statistical Analysis

As most variables were not normally distributed, as assessed using the Kolmogorov–Smirnov test, data were compared using the Mann–Whitney *U*, Kruskal–Wallis, chi-square, or Fisher’s exact test, as appropriate.

The odds ratio (OR) of each covariate was derived from a simple or multiple logistic regression model as an estimated coefficient with a 95% confidence interval (CI). Spearman correlation coefficients were computed between selected predictors. For translational purposes, an optimal cut-off value was used to further dichotomize selected continuous variables using receiver operating characteristic curves with Youden’s J statistic as the best trade-off between sensitivity and specificity, to predict LSCC [34].

Survival curves were plotted using the Kaplan–Meier method and compared using the log-rank test. Optimal cut-off values were used to further dichotomize selected continuous variables using time-dependent receiver operating characteristic curves [35] with Youden’s J statistic as the best trade-off between sensitivity and specificity, to predict the prognoses. Furthermore, hazard ratios (HRs) and their corresponding 95% CIs were calculated. 

Variables with *p* < 0.05 in univariate analyses were entered into multivariate logistic regression models for risk analysis, or Cox regression models for prognosis analysis. Two-tailed *p*-values < 0.05 were considered statistically significant. 

Statistical analyses were conducted using R (version 3.6.1, R Foundation for Statistical Computing, http://www.r-project.org/, accessed on 28 February 2021) and GraphPad Prism for Windows software (version 9.0, GraphPad Software Inc., San Diego, CA, USA). 

## 3. Results

### 3.1. Clinical Characteristics of Patients 

A total of 98 patients were identified, 17 of whom were excluded due to insufficient tissues (*n* = 14) and low biomarker quality (*n* = 3) (Figure 1). Therefore, the overall cohort comprised of 42 patients with LSCC (cases) and 39 patients with NMLL (controls); a total of 81 patients (74 (91%) men and 7 (9%) women) with a median age of 58 years (interquartile age: 51–70 years). Table 1 summarizes the baseline characteristics of the populations in the study. Age ≥ 55 years (unadjusted OR = 8.3; 95% CI: 3.0–22.5; *p* < 0.001) and cigarette smoking (unadjusted OR = 4.5; 95% CI: 1.1–17.8; *p* = 0.03) were significantly associated with a risk of LSCC.

Most patients had early stage (Stage I–II) LSCC (81%) and had undergone transoral laser microsurgery (67%) or RT (10%) as the definitive treatment. 

### 3.2. EBV-Related Biomarkers in the Larynx and Peripheral Blood

The overall laryngeal EBV infection rate was 37% (Table 2). The laryngeal EBV DNA positivity rate was significantly higher in case, compared to control, patients (52% vs. 20%; effect size = 0.70; 95% CI 0.24–1.16; *p* = 0.01); therefore, patients with laryngeal EBV DNA positivity had a greater risk of developing LSCC, compared to those with EBV DNA negativity (unadjusted OR = 4.3; 95% CI: 1.6–11.4; *p* = 0.004). 

The laryngeal nPI for *EBER* was relatively low in both case and control groups, and the staining distribution was equal between the groups (Table 2). Most *EBER* signals were localized to the nuclei of TILs (Figure 2a); however, laryngeal EBV DNA positivity was not related to laryngeal nPI for *EBER* (*r* = 0.11, *p* = 0.34).

Only 30 patients with LSCC had available peripheral blood EBV test data; among these patients, two (7%) and four (13%) had circulating EBV DNA positivity and serum EBV VCA IgA positivity, respectively. Differences in circulating EBV DNA positivity (7% vs. 6%, *p* > 0.99) and EBV VCA IgA positivity (7% vs. 19%, *p* = 0.60) between laryngeal EBV DNA-positive and -negative subgroups were not statistically significant. Therefore, the associations between laryngeal EBV DNA positivity, laryngeal nPI for *EBER*, plasma EBV DNA positivity, and serum EBV VCA IgA positivity were not statistically significant (all *p* > 0.05).

### 3.3. Histological Factors of the Larynx

In the IHC analysis, expression of *p16INK4A*, *BCL-2*, and *MYC* occurred mainly in intratumoral cells (Figure 2b–d), whereas *B2M*, *CD3*, and *CD161* expression occurred mainly in peritumoral (Figure 2e–g) cells. Expression of *BCL-2*, *B2M*, *CD3*, and *CD161* was higher, *MYC* was lower, and *p16INK4a* was the same in LSCC, compared with NMLL (Table 3).

We further dichotomized these biomarkers using receiver operating characteristic curve analyses, with Youden’s J-point as the best trade-off between sensitivity and specificity, to predict LSCC. Expression of *BCL-2* ≥ 68.6% (unadjusted OR = 5.5; 95% CI: 2.0–14.9; *p* = 0.001), *B2M* ≥ 84.3% (unadjusted OR = 8.6; 95% CI: 3.1–23.9; *p* < 0.004), *CD3* ≥ 6.9% (unadjusted OR = 7.2; 95% CI: 2.6–20.2; *p* < 0.001), CD161 ≥ 68.8% (unadjusted OR = 3.5; 95% CI: 1.3–9.4; *p* = 0.01), *MYC* ≤ 0.44% (unadjusted OR = 2.9; 95% CI 1.2–7.1; *p* = 0.02), and *p16INK4a* ≤ 16.8% (unadjusted OR = 6.0; 95% CI 1.6–22.9; *p* = 0.01) were significant risk factors for LSCC.

### 3.4. Relationship between EBV-Related Biomarkers and Clinical and Histological Characteristics in the Overall Cohort

In the overall cohort, primary LSCC was significantly associated with age ≥ 55 years, cigarette smoking, EBV DNA positivity, *BCL-2* expression ≥ 68.6%, *B2M* expression ≥ 84.3%, *CD3* expression ≥ 6.9%, *CD161* expression ≥ 68.8%, *MYC* expression ≤ 0.44%, and *p16INK4a* expression ≤ 16.8% (Table 4). 

Additionally, the male sex was significantly associated with cigarette smoking and alcohol consumption. The associations between age ≥ 55 years and *MYC* expression ≤ 0.44%, *B2M* expression ≥ 84.3%, and *CD3* expression ≥ 6.9% were significant. Furthermore, the relationship between cigarette smoking and alcohol consumption was significant.

EBV DNA positivity was significantly correlated with high *BCL-2* expression (≥68.6%), which also significantly correlated with *B2M* expression ≥ 84.3%, *CD3* expression ≥ 6.9%, and *CD161* expression ≥ 68.8%. Low *p16INK4a* expression (≤16.8%) was significantly associated with *B2M* expression ≥ 84.3% and *CD161* expression ≥ 68.8%. *B2M* expression ≥ 84.3% was significantly associated with *CD3* expression ≥ 6.9%, which also significantly associated with *CD161* expression ≥ 68.8%.

### 3.5. Multi-Factor Modeling of Primary LSCC

To improve the precision of the model, we used a “forced simultaneous entry” approach and a “sign-correct” method for multivariate logistic regression analysis [36]. Variables with a *p* < 0.05 in the univariate analysis were entered into the regression analysis. Age ≥ 55 years (OR = 19.3; 95% CI: 3.2–116.0; *p* = 0.001), cigarette smoking (OR = 36.4; 95% CI: 2.4–554.2; *p* = 0.01), EBV DNA positivity (OR = 39.7; 95% CI: 3.3–478.0; *p* = 0.004), *BCL-2* expression ≥ 68.6% (OR = 6.8; 95% CI: 1.2–38.6; *p* = 0.03), *B2M* expression ≥ 84.3% (OR = 18.1; 95% CI: 2.9–11.6; *p* = 0.002), and *CD161* expression ≥ 68.8% (OR = 7.5; 95% CI: 1.1–50.7; *p* = 0.04) were significant independent risk factors for primary LSCC.

Using this six-factor model, we accurately predicted which patients with laryngeal lesions had primary LSCC via the optimal cut-off value of 3 (area under the receiver operating characteristic curve: 0.87 [95% CI: 0.79–0.96]; *p* < 0.001), with a sensitivity, specificity, positive predictive value, and negative predictive value of 98%, 77%, 82%, and 97%, respectively.

### 3.6. Five Year Prognoses in Patients with LSCC

As of 28 February 2021, the median follow-up time was 68 months (range: 9–94 months). A total of 14 case patients had an LR within the first 5 years after definitive treatment; thus, the five-year LR rate was 34% (95% CI: 16–53%). Two patients had NR (five-year NR rate = 6%; 95% CI: 0–50%) and one had DM (five-year DM rate = 3%; 95% CI: 0–59%); therefore, five-year DFS was 58% (95% CI: 41–71%). Furthermore, two patients had disease-specific deaths (five-year DSS rate = 94%; 95% CI: 75–98%), while another two died of other causes (five-year OS rate = 89%; 95% CI: 72–96%). 

Herein, we further focused on investigating the risk factors associated with five-year LR and DFS rates (Table 5). EBV DNA positivity did not significantly predict five-year LR and DFS rates. However, an *EBER* signal ≥ 0.04% significantly predicted both the five-year LR (unadjusted HR = 6.2; 95% CI: 2.0–18.6; *p* = 0.001) and DFS (unadjusted HR = 7.1; 95% CI: 2.6–19.5; *p* < 0.001) rates using univariate Cox regression models. Furthermore, age ≤ 63 years, TLM, and *CD3* expression ≤ 4.9% also predicted the five-year LR rate, whereas TLM, *BCL-2* expression ≤ 96.0%, *CD3* expression ≤ 4.9%, and *EBER* signals ≥ 0.04% predicted five-year DFS rates.

Using multivariate Cox regression models, both *EBER* signals ≥ 0.04% (adjusted HR: 6.0; 95% CI: 1.9–18.6; *p* = 0.002) and *CD3* expression ≤ 4.9% (adjusted HR = 6.9; 95% CI: 1.9–24.6; *p* = 0.003) independently predicted five-year LR rates; both risk factors significantly predicted five-year LR rates after adjustment for treatment modality (adjusted HR = 4.7 (95% CI: 1.5–15.0) and 5.4 (95% CI: 1.5–19.4), respectively; *p* = 0.01, and 0.01, respectively). Similarly, both *EBER* signals ≥ 0.04% (adjusted HR = 8.6; 95% CI: 2.9–25.3; *p* < 0.001) and *CD3* expression ≤ 4.9% (adjusted HR = 6.6; 95% CI: 1.9–23.6; *p* = 0.004) independently predicted five-year DFS rates; these risk factors remained significant after adjustment for treatment modality (adjusted HR = 7.7 (95% CI: 2.5–24.0) and 5.4 (95% CI: 1.5–19.3), respectively; *p* < 0.001 and 0.01, respectively).

### 3.7. Relationship between EBV-Related Biomarkers and Clinical and Histological Characteristics in Patients with LSCC

In patients with primary LSCC, male sex was significantly associated with cigarette smoking and laryngeal EBV DNA positivity (Table 6). Furthermore, an age ≤ 63 years was associated with alcohol consumption and *p16INK4a* ≤ 81.6%, while cigarette smoking was related to alcohol consumption. As expected, T- and N-stages were associated with LSCC stage. EBV DNA positivity and *EBER* signal ≥ 0.04% were not associated with other tissue biomarkers. *BCL-2* expression ≤ 96.0% correlated significantly with *p16INK4a* and *CD161* expression ≤ 81.6% and ≤ 69.9%, respectively. In addition, *p16INK4a* expression ≤ 81.6% was associated with *CD3* expression ≤ 4.9%.

The overall cohort was further dichotomized into a high-risk group (≥1 risk factor) and a low-risk group (<1 risk factor) to predict five-year LR (sensitivity, 79%; specificity, 94%; positive predictive value, 88%; negative predictive value, 90%). This method performed well in predicting five-year DFS (sensitivity, 76%; specificity, 100%; positive predictive value, 100%; negative predictive value, 52%). The five-year LR rate of the high-risk group was significantly higher than that of the low-risk group (75% (95% CI, 57–86%) vs. 11% (95% CI, 0–49%); *p* < 0.001) (Figure 3a). As expected, the five-year DFS rate of the high-risk group was significantly higher than that of the low-risk group (0% vs. 84% (95% CI, 63–84%); *p* < 0.001) (Figure 3b). However, this model predicted the five-year DSS (83% (95% CI, 47–96%) vs. 100%; *p* = 0.05) and OS (76% (95% CI, 43–92%) vs. 96% (95% CI, 72–99%); *p* = 0.08) rates less efficiently due to relatively low numbers of disease-specific and overall deaths.

## 4. Discussion

In this study, we found that patients with primary LSCC had a higher prevalence of EBV infection, compared to control patients. In addition to age ≥ 55 years and cigarette smoking [37], EBV DNA positivity, *BCL-2* expression ≥ 68.6%, *B2M* expression ≥ 84.3%, and *CD161* expression ≥ 68.8% were independently associated with LSCC. Using this novel model, we accurately predicted the development of LSCC. Moreover, *EBER* signals ≥ 0.04% and *CD3* expression ≤ 4.9% independently predicted five-year LR and DFS rates with or without adjustment for treatment modality. Using this innovative two-factor model, we could also accurately predict the five-year LR and DFS rates of LSCC.

Chronic viral infection is a risk factor for many cancers, such as hepatitis B virus/hepatocellular carcinoma, human papillomavirus/cervical cancer, and EBV/NPC. Quantifying EBV-related biomarker levels in tumor specimens could improve our understanding of the etiology of some laryngeal cancers, and assist in patient selection for treatment and follow-up protocols [38]. However, quantification in FFPE tissue is difficult due to ambiguous uncertainties and the absence of reliable detection methods. Determining the interactions between EBV-related biomarkers and host factors is also challenging. For translation purposes, we developed and validated risk and prognostic models based on EBV-related biomarkers in this case-control study of Taiwanese patients with primary LSCC.

Similar to other studies [8,9,11], we found that EBV infection was significantly more frequent in patients with primary LSCC than in those with NMLL. However, we found that most *EBERs* were detected in the TILs of LSCC. The absence of latent EBV infection in the tumor tissue of LSCC suggests that its multistep nature of development may include “hit-and-run” carcinogenesis in the laryngeal epithelium. Notably, EBV DNA positivity was positively correlated with *BCL-2* expression, an anti-apoptotic protein in tumor cells. *Wp*, a viral promoter located within *BamHI W* repeats of the EBV genome, activates the gene encoding the viral *BCL-2* homolog *BHRF1*, thereby increasing apoptosis resistance [39]. Moreover, oncogenic EBV latent membrane protein 1 directly upregulates *BCL-2* in NPC [40], potentially also contributing to LSCC [11]. 

Furthermore, we found that *BCL-2* overexpression in LSCC was common (80%). *BCL-2* interacts with *Hsp90β*, and may be involved in the anti-apoptotic progression of LSCC [41]. *BCL-2* overexpression appears to be associated with a complete response to induction chemotherapy [16] and worse prognoses due to tumoral radioresistance [42]. Although *BCL-2* overexpression might not contribute to the prognostic significance of LSCC development [43], we found that both EBV DNA positivity and *BCL-2* overexpression in the larynx were independently associated with a risk of LSCC. Our results also indicate that *BCL-2* expression is not related to tumor aggressiveness and prognosis after a single surgical treatment [42].

Regarding therapeutic prognoses, we found that *EBER* signals ≥ 0.04% and *CD3* expression ≤ 4.9% were important prognostic factors for five-year LR and DFS in patients with LSCC. In our literature review, only a few patients with LSCC had positive *EBER* signals in the larynx [39,44]. Although positive *EBER* signals in LSCC tissues are considered poorer predictors of head and neck squamous cell carcinoma (HNSCC) [39], a small LSCC sample size (two and six cases of positive and negative EBER signals, respectively) is difficult to draw conclusions from. Therefore, this study was the first to identify a high laryngeal *EBER* signal as a poorer prognostic factor in LSCC. Since most *EBER* signals presented the peritumor microenvironment in our patients, we presumed that *EBERs* could induce the initial transformation of epithelial cells [45], and trigger cancer-related inflammation via the RIG-I pathway to promote tumor development and growth [46]; rather than downregulate B2M expression to evade T cell-mediated cytotoxic immune responses [47], as according to previous NPC studies. Moreover, EBV-infected TILs of residual laryngeal tissue following curative treatment may increase the five-year LR risk via *BHRF1*-mediated apoptosis resistance [48]. Accordingly, mechanistic studies are warranted to confirm that EBV involves the *EBER*-mediated process of LSCC recurrence.

Circulating EBV DNA is a robust biomarker for EBV-associated NPC [49] and lymphoma [50]. EBV VCA IgA positivity indicates previous repeated EBV infections or frequent reactivation of latent EBV in B cells [51]. Patients with LSCC might have a higher positive rate of EBV VCA IgA than healthy controls [52]. Therefore, both circulating EBV DNA and VCA IgA could be potential biomarkers for EBV-positive LSCC. In this study, neither circulating EBV DNA positivity nor serum EBV VCA IgA positivity was associated with laryngeal EBV DNA positivity and EBER nPI. Furthermore, the low positive rates of these peripheral blood biomarkers may limit their application in patients with LSCC. 

Laryngeal cancer usually develops over many years, and approximately 80% of patients with LSCC are 60 years or older at first discovery [53]. In this study, an age ≥ 55 years was associated with *B2M* expression ≥ 84.3%, and both were independent risk factors for LSCC development. Recently, *B2M* was validated as a key reference gene in laryngeal and hypopharyngeal cancers, highlighting its suitability for investigating target gene expression [54]. Interestingly, its expression in patients with EBV-positive cancer was significantly higher than that in patients with EBV-negative cancer [55]. The *B2M* protein forms the light chain of the class I major histocompatibility complex, and is important for antigen recognition by cytotoxic T cells [56]. Overexpression of *B2M* is found in solid tumors and blood-borne malignancies, and is associated with advanced disease and poor prognosis [57]. Although *B2M* expression was not associated with LSCC stage and prognosis, it was an independent risk factor for LSCC development.

Furthermore, cigarette smoking is strongly associated with an increased risk of LSCC in a dose-response and time-response manner [58]. Although significant smoking trend reductions have led to the declined global burden of LSCC in developed countries, smoking continues to trend upward in low socioeconomic countries, potentially inducing an increased burden of LSCC in the future [59]. Notably, smoking prevention programs conducted in health care settings and interventions, including enhancement of interpersonal communication and support strategies, can be effective in preventing smoking behavior for three months to four years in children and adolescents [60]. However, the latest meta-analysis indicated that findings which suggest that reduction-to-quit is more effective than no treatments were inconclusive and of low certainty [61]. Nevertheless, the risk of developing LSCC can be reduced by smoking cessation for ≥ 15 years [58].

*CD161* is expressed on natural killer (NK) cells and subpopulations of T lymphocytes, such as invariant natural killer T cells (iNKT), CD4^+^CD161^+^ T cells, and CD8^+^CD161^+^ T cells [62]. These cells may play significant roles in tumor development [63], and may regulate the immune response in the tumor microenvironment [64]. However, the effect of *CD161* expression on tumor development remains unclear. Patients with LSCC had a significantly lower percentage of circulating iNKT cells, compared to healthy controls [65], whereas circulating CD4^+^CD161^+^ T cells, representing a memory T cell population, significantly increased in cancer patients [66]. Notably, CD8^+^CD161^+^ T cell-specific EBV infections might express high levels of anti-apoptotic molecules to survive hostile inflammatory conditions, also involving the pathogenesis of tumor tissues [67]. Furthermore, we found that *CD161* overexpression in LSCC was not uncommon, and correlated with *BCL-2* overexpression and low *p16INK4a* expression. However, both *CD161* and *BCL-2* overexpression were independent risk factors for LSCC development. 

In patients with LSCC, low *CD3* expression was associated with a poor prognosis. Similarly, a high infiltration of CD3^+^ T lymphocytes was associated with a significantly better prognosis in patients with HNSCC [68]. In patients with HNSCC, low *CD3* mRNA levels have been found to be worse predictors of 8-year DFS [69]. Therefore, dysfunction of the immune response in removing the oncogenic EBV infection may increase the risk of cancer progression. Although we did not find a significant association between *CD3* expression and EBV infection in the larynx, patients with LSCC and EBV infection may have a lower proportion of late activated T lymphocytes in the peripheral blood compared to patients without EBV infection [70]. Therefore, investigating peripheral blood T lymphocytes may help us to comprehensively understand the possible mechanisms of LR and relapse in EBV-related LSCC.

Our study had some limitations. First, when considering high EBV DNA positivity, false positivity is a real risk; PCR, which was used for EBV DNA detection, is extremely sensitive, and the EBER signal was minimal. However, we applied strict anti-contamination measures in the laboratory to reduce the risk of contamination [27]. Second, in some patients, insufficient tissue and low-quality biomarkers limited the sample size; despite the use of TMAs to reduce the lesion volume required for comprehensive comparisons. Nevertheless, larger sample sizes may provide only modest benefits in complex statistical analyses. 

## 5. Conclusions

Taken together, our study is the most comprehensive analysis to date of the clinical and histological manifestations, and etiology of primary LSCCs and five-year prognoses following definitive treatment. We identified EBV DNA positivity, older age, cigarette smoking, and higher expression of *BCL-2*, *B2M*, and *CD161* as risk factors for primary LSCC. Furthermore, we identified high *EBER* signals and low *CD3* expression as independent predictors of five-year LR and DFS. Early prediction will enable the development of better treatments and follow-up strategies. Furthermore, as understanding the etiology of LSCC and reducing its LR rate is paramount, our data provide significant information that may assist in the accomplishment of these goals.

## Figures and Tables

**Figure 1 cancers-13-01741-f001:**
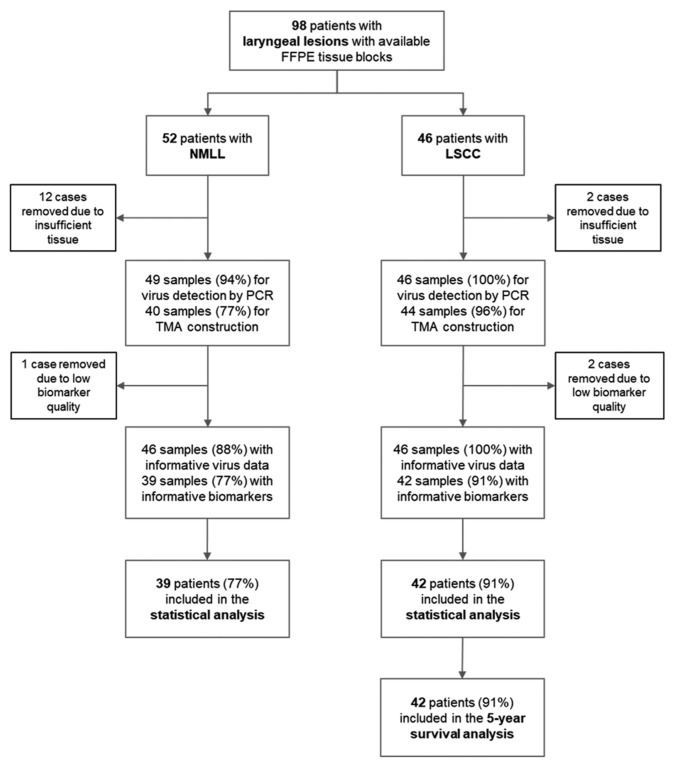
REMARK diagram. There were 94 patients with laryngeal lesions with available formalin-fixed paraffin-embedded (FFPE) tissue blocks. Seventeen cases with inadequate tissue and low-quality biomarkers were excluded. Therefore, 42 patients with laryngeal squamous cell carcinoma (LSCC) and 39 sex-matched control patients with a non-malignant lesion of the larynx (NMLL) were included in the final statistical analyses.

**Figure 2 cancers-13-01741-f002:**
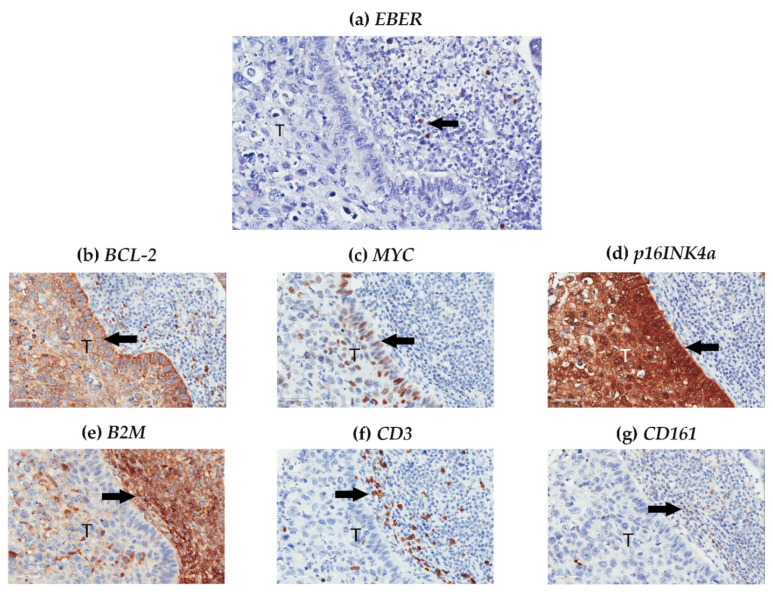
Evaluation of Epstein–Barr virus-encoding RNA (*EBER*), *BCL-2*, *MYC*, *p16INK4a*, *B2M*, *CD3*, and *CD161* in a representative case with laryngeal squamous cell carcinoma. (**a**) *EBER* signals (arrow) were detected using in situ hybridization in the nuclei of tumor-infiltrating lymphocytes. Expression of tumor markers, including (**b**) *BCL-2*, (**c**) *MYC*, and (**d**) *p16INK4a* (arrows), occurred predominantly in intratumoral cells, whereas expression of immunological markers, including (**e**) *B2M*, (**f**) *CD3*, and (**g**) *CD161* (arrows), occurred mainly in peritumoral cells. Original magnification 40×.

**Figure 3 cancers-13-01741-f003:**
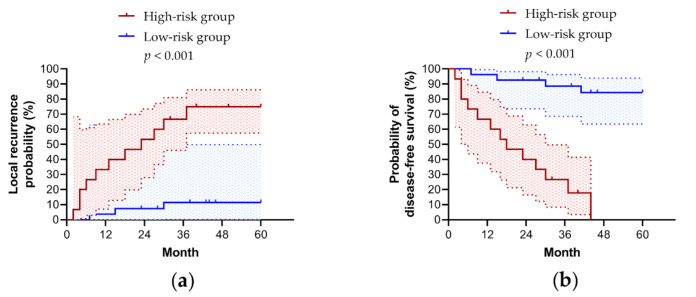
Kaplan-Meier curves of the predictive models of five-year prognoses. Comparison of (**a**) five-year local recurrence rates and (**b**) five-year disease-free survival rates between high-risk and low-risk groups using log-rank tests.

**Table 1 cancers-13-01741-t001:** Clinicopathological characteristics and treatments.

Characteristic	Overall	Patients with LSCC	Patients with NMLL	*p*-Value ^a^
**Clinical Characteristics**				
Patients	*n* = 81	*n* = 42	*n* = 39	
Male, *n* (%)	74 (91)	38 (91)	36 (92)	>0.99
Female, *n* (%)	7 (9)	4 (9)	3 (8)	
Age (years), median (range)	58 (51–70)	**64 (58–74)**	**52 (37–58)**	**<0.001**
Age ≥ 55 years, *n* (%) ^b^	45 (56)	**33 (79)**	**12 (31)**	**<0.001**
Age < 55 years, *n* (%) ^b^	36 (44)	**9 (21)**	**27 (69)**	
Cigarette smoker, *n* (%)	69 (85)	**39 (93)**	**29 (74)**	**0.03**
Never smoker, *n* (%)	12 (15)	**3 (7)**	**10 (26)**	
Alcohol drinker, *n* (%)	48 (59)	27 (64)	21 (54)	0.37
Never drinker, *n* (%)	33 (41)	15 (36)	18 (46)	
**Pathological Characteristics**				
Polyps, *n* (%)	–	–	18 (46)	–
Leukoplakia, *n* (%)	–	–	21 (54)	–
T1–2, *n* (%)	–	34 (81)	–	–
T3–4, *n* (%)	–	8 (19)	–	–
N0, *n* (%)	–	40 (95)	–	–
N1–2, *n* (%)	–	2 (5)	–	–
Stage I–II, *n* (%)	–	34 (81)	–	–
Stage III–IV, *n* (%)	–	8 (19)	–	–
**Treatment Modalities**				
Transoral laser microsurgery, *n* (%)	–	28 (67)	–	–
Total laryngectomy, *n* (%)		3 (7)		–
RT, *n* (%)	–	4 (10)	–	–
CCRT, *n* (%)	–	3 (7)	–	–
Surgery + RT, *n* (%)	–	3 (7)	–	–
Surgery + CCRT, *n* (%)		1 (2)	–	–
Single-modality treatment, *n* (%)		31 (74)	–	–
Multiple-modality treatment, *n* (%)	–	11 (26)	–	–

Abbreviations: CCRT, concurrent chemoradiotherapy; LSCC, laryngeal squamous cell carcinoma; NMLL, non-malignant laryngeal lesion; RT, radiotherapy. ^a^ Data were compared between LSCC and NMLL subgroups using the Mann-Whitney *U* test or Fisher’s exact test, as appropriate. ^b^ The optimal cut-off value using receiver operating characteristic curves with Youden’s *J*-point as the best trade-off between sensitivity and specificity, to predict LSCC. Significant *p*-values are marked in bold.

**Table 2 cancers-13-01741-t002:** EBV status in laryngeal tissue and peripheral blood.

Characteristics	Overall	Patients with LSCC	Patients with NMLL	*p*-Value ^a^
**Laryngeal Tissue**				
Patients	*n* = 81	*n* = 42	*n* = 39	
EBV DNA positivity, *n* (%)	30 (37)	**22 (52)**	**8 (20)**	**0.01**
EBV DNA negativity, *n* (%)	51 (63)	**20 (48)**	**31 (80)**	
nPI for *EBER*, (%), median (range)	0.02 (0.01–0.06)	**0.03 (0.01–0.05)**	**0.02 (0.01–0.07)**	**<0.001**
*EBER* signal ≥ 0.02%, *n* (%) ^b^	50 (62)	28 (67)	22 (56)	0.37
*EBER* signal < 0.02%, *n* (%) ^b^	31 (62)	14 (33)	17 (44)	
**Peripheral Blood**				
Patients	–	*n* = 30	–	
Circulating EBV DNA positivity, *n* (%)	–	2 (7)	–	–
Circulating EBV DNA negativity, *n* (%)	–	28 (93)	–	–
EBV-VCA IgA positivity, *n* (%)	–	4 (13)	–	–
EBV VCA IgA negativity, *n* (%)	–	26 (87)	–	–

Abbreviations: *EBER*, Epstein–Barr encoding region; EBV, Epstein–Barr virus; LSCC, laryngeal squamous cell carcinoma; NMLL, non-malignant laryngeal lesion; VCA, viral capsid antigen; PI, positivity index. ^a^ Data were compared between LSCC and NMLL subgroups using the Mann-Whitney *U* test or Fisher’s exact test, as appropriate. ^b^ The optimal cut-off value using receiver operating characteristic curves with the Youden’s *J*-point as the best trade-off between sensitivity and specificity, to predict LSCC. Significant results with their *p*-values are marked in bold.

**Table 3 cancers-13-01741-t003:** Histological factors of the larynx.

Characteristics	Overall	Patients with LSCC	Patients with NMLL	*p*-Value ^a^
**Tumor-Related Biomarkers in the Larynx**				
Patients	*n* = 81	*n* = 42	*n* = 39	
cPI for *BCL-2*, (%), median (range)	82.9 (43.3–98.2)	**86.7 (76.8–98.7)**	**61.5 (35.5–94.2)**	**0.01**
*BCL-2* expression ≥ 68.6%, *n* (%) ^b^	30 (37)	**34 (81)**	**17 (44)**	**0.001**
*BCL-2* expression < 68.6%, *n* (%) ^b^	51 (63)	**8 (19)**	**22 (56)**	
nPI for *MYC*, (%), median (range)	0.44 (0.24–0.56)	**0.38 (0.17–0.53)**	**0.50 (0.32–0.67)**	**0.03**
*MYC* expression ≤ 0.44%, *n* (%) ^b^	44 (54)	**28 (67)**	**16 (41)**	**0.03**
*MYC* expression > 0.44%, *n* (%) ^b^	37 (46)	**14 (33)**	**23 (59)**	
cPI for *p16INK4a*, (%), median (range)	49.8 (20.3–93.8)	45.3 (11.5–92.4)	49.9 (25.0–96.3)	0.21
*p16INK4a* expression ≤ 16.8%, *n* (%) ^b^	17 (21)	**14 (33)**	**3 (8)**	**0.01**
*p16INK4a* expression > 16.8%, *n* (%) ^b^	64 (79)	**28 (67)**	**36 (92)**	
**Host Mucosal Immune-Related Biomarkers in the Larynx**				
cPI for *B2M*, (%), median (range)	82.9 (68.2–96.3)	**92.5 (79.5–97.6)**	**74.1 (55.9–84.0)**	**0.001**
*B2M* expression ≥ 84.3%, *n* (%) ^b^	37 (46)	**29 (69)**	**8 (21)**	**<0.001**
*B2M* expression < 84.3%, *n* (%) ^b^	44 (54)	**13 (31)**	**31 (79)**	
cPI for *CD3*, (%), median (range)	11.3 (3.8–27.3)	**20.5 (9.1–32.8)**	**6.3 (2.0–15.4)**	**0.001**
*CD3* expression ≥ 6.9%, *n* (%) ^b^	51 (63)	**35 (83)**	**16 (41)**	**<0.001**
*CD3* expression < 6.9%, *n* (%) ^b^	30 (37)	**7 (17)**	**23 (59)**	
cPI for *CD161*, (%), median (range)	47.9 (14.7–76.9)	**59.8 (23.5–80.9)**	**32.0 (6.7–67.3)**	**0.004**
*CD161* expression ≥ 68.8%, *n* (%) ^b^	28 (35)	**20 (48)**	**8 (21)**	**0.02**
*CD161* expression < 68.8%, *n* (%) ^b^	53 (65)	**22 (52)**	**31 (80)**	

Abbreviations: *EBER*, Epstein–Barr encoding region; EBV, Epstein–Barr virus; LSCC, laryngeal squamous cell carcinoma; NMLL, non-malignant laryngeal lesion; VCA, viral capsid antigen; PI, positivity index. ^a^ Data were compared between LSCC and NMLL subgroups using the Mann-Whitney *U* test or Fisher’s exact test, as appropriate. ^b^ The optimal cut-off value using receiver operating characteristic curves with Youden’s *J*-point as the best trade-off between sensitivity and specificity, to predict LSCC. Significant results with their *p*-values are marked in bold.

**Table 4 cancers-13-01741-t004:** Spearman’s correlations between clinical characteristics, laryngeal EBV-related biomarkers, and laryngeal histological characteristics in the overall cohort.

Characteristics	LSCC	Male Sex	Age ≥ 55 Years	Cigarette Smoking	Alcohol Consumption	EBV DNA Positivity	*EBER* Signal ≥ 0.02%	*BCL-2* Expression ≥ 6 8.6%	*MYC* Expression ≤ 0.44%	*p16INK4a* Expression ≤ 16.8%	*B2M* Expression ≥ 84.3%	*CD3* Expression ≥ 6.9%	*CD161* Expression ≥ 68.8%
LSCC	–												
Male sex	−0.03	–											
Age ≥ 55 years	**0.48 ^c^**	−0.01	–										
Cigarette smoking	**0.25 ^a^**	**0.46 ^c^**	0.08	–									
Alcohol consumption	0.11	**0.28 ^a^**	−0.08	**0.32 ^b^**	–								
EBV DNA positivity	**0.44 ^c^**	−0.20	0.18	0.05	−0.04	–							
*EBER* signal ≥ 0.02%	0.11	0.12	0.22	0.01	−0.03	−0.12	–						
*BCL-2* expression ≥ 68.6%	**0.39 ^c^**	−0.05	0.09	−0.06	0.04	**0.26 ^a^**	0.03	–					
*MYC* expression ≤ 0.44%	**0.26 ^a^**	−0.19	**0.33 ^b^**	0.14	−0.16	0.14	0.15	0.07	–				
*p16INK4a* expression ≤ 16.8%	**0.32 ^b^**	−0.06	−0.09	0.14	0.12	0.08	−0.03	0.14	0.05	–			
*B2M* expression ≥ 84.3%	**0.49 ^c^**	0.02	**0.22 ^a^**	0.06	0.11	0.14	0.06	**0.24 ^a^**	0.14	**0.32 ^b^**	–		
*CD3* expression ≥ 6.9%	**0.44 ^c^**	−0.05	**0.24 ^a^**	0.08	0.15	0.20	0.13	**0.52 ^c^**	0.17	0.21	**0.40 ^c^**	–	
*CD161* expression ≥ 68.8%	**0.29 ^a^**	−0.05	0.02	0.11	−0.03	0.06	0.20	**0.24 ^a^**	0.20	**0.39 ^c^**	0.17	**0.29 ^b^**	–

Abbreviations: *EBER*, Epstein–Barr encoding region; EBV, Epstein–Barr virus; LSCC, laryngeal squamous cell carcinoma. Data are summarized as *r*-vales. Significant results are marked in bold: ^a^
*p* ≥ 0.01 to < 0.05, ^b^
*p* ≥ 0.001 to < 0.01, and ^c^
*p* < 0.001.

**Table 5 cancers-13-01741-t005:** Cox regression models of variables of interest for predicting five-year prognosis in patients with LSCC.

Characteristics	5-Year Local Recurrence					5-Year Disease-Free Survival				
	Cut-Off Value	Univariate Models		Multivariate Models		Cut-Off Value	Univariate Models		Multivariate Models	
		HR (95% CI)	*p*-Value	HR (95% CI)	*p*-Value		HR (95% CI)	*p*-Value	HR (95% CI)	*p*-Value
Male sex	yes	0.5 (0.1–2.5)	0.43			yes	0.7 (0.2–3.1)	0.65		
Age, years	≤63	**3.2 (1.0** **–10** **.3** **)**	**0.048**	–	NS	≤63	1.8 (0.7–4.8)	0.23		
Cigarette smoking	yes	23.2 (0.1–82375.3)	0.45			yes	23.6 (0.1–29976.8)	0.39		
Alcohol drinking	yes	2.2 (0.6–7.9)	0.23			yes	2.0 (0.6–6.1)	0.23		
T-stage	≤2	30.7 (0.2–6190.3)	0.21			≤3	32.8 (0.3–3521.6)	0.14		
N-stage	0	22.0 (0–4.4 × 10^5^)	0.54			≤0	22.3 (0.1–1.3×10^5^)	0.48		
pStage	≤2	30.7 (0.2–6190.3)	0.21			≤2	32.8 (0.3–3521.6)	0.14		
Treatment	TLM	**8.3 (1.1–63.8)**	**0.04**	–	NS	TLM	**5.6 (1.3–24.9)**	**0.02**	–	NS
EBV DNA positivity	yes	0.56 (0.2–1.64)	0.28			yes	0.4 (0.1–1.1)	0.08		
*EBER* signal, %	≥0.04	**6.2 (2.0** **–18.6)**	**0.001**	**6.0 (1.9–18.6)**	**0.002**	≥0.04	**7.1 (2.6** **–19.5)**	**<0.001**	**8.6 (2.9–25.3)**	**<0.001**
*BCL-2* expression, %	≤96.0	2.9 (0.8–10.6)	0.10			≤96.0	**3.9 (1.1–13.6)**	**0.03**	–	NS
*MYC* expression, %	≤0.50	3.1 (0.7–13.8)	0.14			≤0.50	4.2 (1.0–18.6)	0.06		
*p16INK4a* expression, %	≤81.6	1.1 (0.4–3.4)	0.83			≤81.6	2.1 (0.7–6.0)	0.16		
*B2M* expression, %	≤92.7	3.5 (1.0–12.7)	0.05			≤93.8	2.0 (0.7–5.7)	0.19		
*CD3* expression, %	≤4.9	**7.2 (2.2–1209)**	**0.001**	**6.9 (1.9–24.6)**	**0.003**	≤4.9	**6.7 (2.1–21.8)**	**0.001**	**6.6 (1.9–23.6)**	**0.004**
*CD161* expression, %	≤69.9	2.4 (0.7–7.5)	0.15			≤73.1	1.8 (0.6–5.2)	0.26		

Abbreviations: *EBER*, Epstein–Barr encoding region; EBV, Epstein–Barr virus; HR, hazard ratio; LSCC, laryngeal squamous cell carcinoma; NS, not significant. Significant results with their *p*-values are marked in bold.

**Table 6 cancers-13-01741-t006:** Spearman’s correlations between clinical characteristics, laryngeal EBV-related biomarkers, and laryngeal histological characteristics in patients with LSCC.

Characteristics	Male Sex	Age ≤ 63 years	Cigarette Smoking	Alcohol Consumption	T-Stage ≤ 2	N-Stage = 0	Stage ≤ 2	Laryngeal EBV DNA Positivity	*EBER* Signal ≥ 0.04%	*BCL-2* Expression ≤ 96.0%	*MYC* Expression ≤ 0.50%	*p16INK4a* Expression ≤ 81.6%	*B2M* Expression ≤ 92.7%	*CD3* Expression ≤ 4.9%	*CD161* Expression ≤ 69.9%
Male sex	–														
Age ≤63 years	−0.02	–													
Cigarette smoking	**0.54 ^a^**	0.08	–												
Alcohol consumption	0.27	**0.31 ^b^**	**0.37 ^b^**	–											
T-stage ≤ 2	0.13	−0.10	0.25	−0.02	–										
N-stage = 0	−0.07	−0.01	−0.06	0.07	0.20	–									
Stage ≤ 2	0.02	−0.15	0.10	0.02	**0.44 ^c^**	**0.46 ^c^**	–								
Laryngeal EBV DNA positivity	**−0.34 ^b^**	−0.15	−0.11	0.01	−0.29	−0.24	−0.27	–							
*EBER* signal ≥ 0.04%	0.03	0.24	0.18	0.14	0.06	0.14	**0.31 ^b^**	−0.18	–						
*BCL-2* expression ≤ 96.0%	0.06	0.20	0.15	0.20	0.07	−0.18	0.22	−0.09	0.20	–					
*MYC* expression ≤ 0.50%	−0.22	−0.29	0.01	−0.18	0.20	0.09	0.20	0.02	0.20	0.08	–				
*p16INK4a* expression ≤ 81.6%	0.05	**0.44 ^c^**	0.13	0.16	0.11	0.03	0.07	−0.23	0.12	**0.36 ^b^**	0.25	–			
*B2M* expression ≤ 92.7%	0.03	0.01	−0.07	0.12	0.06	0.02	0.05	0.20	0.15	0.13	0.12	−0.21	–		
*CD3* expression ≤ 4.9%	0.12	0.09	0.10	−0.03	0.11	0.08	0.18	−0.20	0.09	0.30	0.09	**0.32 ^b^**	0.19	–	
*CD161* expression ≤ 69.9%	0.03	0.01	−0.07	0.02	−0.04	0.02	0.05	0.01	0.15	**0.32 ^b^**	−0.20	−0.30	0.42 ^c^	0.19	–

Abbreviations: *EBER*, Epstein–Barr encoding region; EBV, Epstein–Barr virus; LSCC, laryngeal squamous cell carcinoma. Data are summarized as *r*-vales. Significant results are marked in bold: ^a^
*p* ≥ 0.01 to <0.05, ^b^
*p* ≥ 0.001 to < 0.01, and ^c^
*p* < 0.001.

## Data Availability

The data presented in this study are available upon request from the corresponding author. The data are not publicly available due to ethical restrictions.

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
