# Peer review of "Effects of Epstein-Barr Virus Infection on the Risk and Prognosis of Primary Laryngeal Squamous Cell Carcinoma: A Hospital-Based Case-Control Study in Taiwan"

_cancers, 2021, doi:10.3390/cancers13071741_

Round 1

Reviewer 1 Report

Please follow REMARK guidelines.

After this adjustment I will be able to further revise the manuscript. This change will improve overall quality and making results comparable in future systematic review and meta-analysis. 

Author Response

Reviewer 1’s comments:

Please follow REMARK guidelines.

After this adjustment I will be able to further revise the manuscript. This change will improve overall quality and making results comparable in future systematic review and meta-analysis.

REPLY. Thank you very much for this in-depth suggestion and encouraged comment. We’ve revised this manuscript followed the recommendations for prognostic studies of tumor biomarkers (REMARK) guidelines. For your convenience, we summarize the reported items using the REMAK checklist.

Item to be reported

Page no.

INTRODUCTION

1

State the marker examined, the study objectives, and any pre-specified hypotheses. 

1

MATERIALS AND METHODS

Patients

2

Describe the characteristics (e.g., disease stage or co-morbidities) of the study patients, including their source and inclusion and exclusion criteria. 

1, 5, 6

3

Describe treatments received and how chosen (e.g., randomized or rule-based). 

3

Specimen characteristics

4

Describe type of biological material used (including control samples) and methods of preservation and storage.

3

Assay methods

5

Specify the assay method used and provide (or reference) a detailed protocol, including specific reagents or kits used, quality control procedures, reproducibility assessments, quantitation methods, and scoring and reporting protocols. Specify whether and how assays were performed blinded to the study endpoint.

3–5

Study design

6

State the method of case selection, including whether prospective or retrospective and whether stratification or matching (e.g., by stage of disease or age) was used. Specify the time period from which cases were taken, the end of the follow-up period, and the median follow-up time. 

1

7

Precisely define all clinical endpoints examined.

3

8

List all candidate variables initially examined or considered for inclusion in models.

5,6

9

Give rationale for sample size; if the study was designed to detect a specified effect size, give the target power and effect size.

4, 5

Statistical analysis methods

10

Specify all statistical methods, including details of any variable selection procedures and other model-building issues, how model assumptions were verified, and how missing data were handled.

5

11

Clarify how marker values were handled in the analyses; if relevant, describe methods used for cutpoint determination.

5

RESULTS

Data

12

Describe the flow of patients through the study, including the number of patients included in each stage of the analysis (a diagram may be helpful) and reasons for dropout. Specifically, both overall and for each subgroup extensively examined report the numbers of patients and the number of events.

5,6

13

Report distributions of basic demographic characteristics (at least age and sex), standard (disease-specific) prognostic variables, and tumor marker, including numbers of missing values.

5

Analysis and presentation

14

Show the relation of the marker to standard prognostic variables.

6–11

15

Present univariable analyses showing the relation between the marker and outcome, with the estimated effect (e.g., hazard ratio and survival probability). Preferably provide similar analyses for all other variables being analyzed. For the effect of a tumor marker on a time-to-event outcome, a Kaplan-Meier plot is recommended.

12

16

For key multivariable analyses, report estimated effects (e.g., hazard ratio) with confidence intervals for the marker and, at least for the final model, all other variables in the model.

13, 14

17

Among reported results, provide estimated effects with confidence intervals from an analysis in which the marker and standard prognostic variables are included, regardless of their statistical significance.

5–15

18

If done, report results of further investigations, such as checking assumptions, sensitivity analyses, and internal validation.

15

DISCUSSION

19

Interpret the results in the context of the pre-specified hypotheses and other relevant studies; include a discussion of limitations of the study.

15–17

20

Discuss implications for future research and clinical value.

17

Special note:

Modified text, Page 9, lines 89-91

‘… accordance with the World Medical Association Declaration of Helsinki.’

à

‘… accordance with the World Medical Association Declaration of Helsinki. This study followed the recommendations for prognostic studies of tumor biomarkers (REMARK) [21].’

Modified text, Page 5, line 231

‘… tissues (n = 14) and low biomarker quality (n = 3). Therefore, the overall cohort …’

-->

‘… tissues (n = 14) and low biomarker quality (n = 3) (Figure 1). Therefore, the overall cohort …’

Modified text, Page 6, lines 238-243

Add Figure 1.

à

Figure 1. REMARK diagram. There were 94 patients with laryngeal lesions with available formalin-fixed paraffin-embedded (FFPE) tissue blocks. Seventeen cases with inadequate tissue and low-quality biomarkers were excluded. Therefore, 42 patients with laryngeal squamous cell carcinoma (LSCC) and 39 sex-matched control patients with a non-malignant lesion of the larynx (NMLL) were included in the final statistical analyses.

Modified text, Page 7, line 264

‘… were localized to the nuclei of TILs (Figure 1a); however, laryngeal EBV DNA positivity …’

-->

‘… were localized to the nuclei of TILs (Figure 2a); however, laryngeal EBV DNA positivity …’

Modified text, Page 8, line 268

Figure 1. Evaluation of Epstein-Barr virus-encoding RNA (EBER), BCL-2, MYC, p16INK4a, B2M, …’

-->

Figure 2. Evaluation of Epstein-Barr virus-encoding RNA (EBER), BCL-2, MYC, p16INK4a, B2M, …’

Modified text, Page 9, lines 295-296

‘… intratumoral cells (Figure 1b–1d), whereas B2M, CD3, and CD161 expression occurred mainly in peritumoral (Figure 1e–1g) cells. Expression of BCL-2, B2M, CD3, and …’

-->

‘… intratumoral cells (Figure 2b–2d), whereas B2M, CD3, and CD161 expression occurred mainly in peritumoral (Figure 2e–2g) cells. Expression of BCL-2, B2M, CD3, and …’

Modified text, Page 12, lines 354-356

‘… As of February 28, 2021, 14 case patients had an LR within the first 5 years after definitive treatment; thus, the 5-year LR rate was 34% (95% CI: 16–53%). …’

-->

‘… As of February 28, 2021, the median follow-up time was 68 months (range: 9‒94 months). A total of 14 case patients had an LR within the first 5 years after definitive treatment; thus, the 5-year LR rate was 34% (95% CI: 16–53%). …’

Modified text, Page 15, lines 404-406

‘… vs. 11% [95% CI, 0%–49%]; p < 0.001) (Figure 2a). As expected, the 5-year DFS rate of the high-risk group was significantly higher than that of the low-risk group (0% vs. 84% [95% CI, 63%–84%]; p < 0.001) (Figure 2b). However, this model predicted the 5-year DSS (83% …’

-->

‘… vs. 11% [95% CI, 0%–49%]; p < 0.001) (Figure 3a). As expected, the 5-year DFS rate of the high-risk group was significantly higher than that of the low-risk group (0% vs. 84% [95% CI, 63%–84%]; p < 0.001) (Figure 3b). However, this model predicted the 5-year DSS (83% …’

Modified text, Page 15, line 410

Figure 2. Kaplan-Meier curves of the predictive models of 5-year prognoses. …’

-->

Figure 3. Kaplan-Meier curves of the predictive models of 5-year prognoses. …’

Reviewer 2 Report

I really appreciate the author's effort to try to find out new prognosticators for LSCC. Even if the cohort of patients is small and comes from a single institution, the article is well written and the statistical analysis is complete and well presented.

Author Response

Reviewer 2’s comments:

I really appreciate the author's effort to try to find out new prognosticators for LSCC. Even if the cohort of patients is small and comes from a single institution, the article is well written and the statistical analysis is complete and well presented.

REPLY. We appreciate these encouraging comments. We do our best and believe that this paper offers insight into the associations between Epstein-Barr virus infection and laryngeal squamous cell carcinoma. The results of this study highlight promising future research pathways through which precision medicine protocols may be developed for patients with laryngeal squamous cell carcinoma. Thank you very much!

Reviewer 3 Report

In this retrospective case-control study the authors assessed the prevalence of EBV infection in patients with laryngeal squamous cell carcinoma.

The study is interesting and well-conducted

Author Response

Reviewer 3’s comments:

In this retrospective case-control study the authors assessed the prevalence of EBV infection in patients with laryngeal squamous cell carcinoma. The study is interesting and well-conducted

REPLY. We appreciate your encouraging comments. We do our best and believe that this paper offers insight into the associations between Epstein-Barr virus infection and laryngeal squamous cell carcinoma. The results of this study highlight promising future research pathways through which precision medicine protocols may be developed for patients with laryngeal squamous cell carcinoma. Thank you very much!
